# Design, Synthesis, and Preliminary Evaluation for Ti-Mo-Zr-Ta-Si Alloys for Potential Implant Applications

**DOI:** 10.3390/ma14226806

**Published:** 2021-11-11

**Authors:** Madalina Simona Baltatu, Mihaela Claudia Spataru, Liliana Verestiuc, Vera Balan, Carmen Solcan, Andrei Victor Sandu, Victor Geanta, Ionelia Voiculescu, Petrica Vizureanu

**Affiliations:** 1Department of Technologies and Equipments for Materials Processing, Faculty of Materials Science and Engineering, Gheorghe Asachi Technical University of Iaşi, Blvd. Mangeron, No. 51, 700050 Iasi, Romania; cercel.msimona@yahoo.com; 2Public Health Department, Faculty of Veterinary Medicine, “Ion Ionescu de la Brad” University of Life Sciences, 3 Mihail Sadoveanu Alley, 700490 Iasi, Romania; spatarufmv@yahoo.com; 3Biomedical Sciences Department, Faculty of Medical Bioengineering, Grigore T. Popa University of Medicine and Pharmacy, 9-13 Kogalniceanu Street, 700454 Iasi, Romania; liliana.verestiuc@bioinginerie.ro (L.V.); balan.vera@umfiasi.ro (V.B.); 4Preclinics Department, Faculty of Veterinary Medicine, “Ion Ionescu de la Brad” University of Life Sciences, 3 Mihail Sadoveanu Alley, 700490 Iasi, Romania; carmensolcan@yahoo.com; 5Romanian Inventors Forum, Str. Sf. P. Movila 3, 700089 Iasi, Romania; 6Engineering and Management of Metallic Material Processing Department, Faculty of Materials Science and Engineering, University Politehnica of Bucharest, 313 Spl Independentei, 060042 Bucharest, Romania; victorgeanta@yahoo.com; 7Quality Engineering and Industrial Technologies Department, Faculty of Industrial Engineering and Robotics, University Politehnica of Bucharest, 313 Spl Independentei, 060042 Bucharest, Romania; ioneliav@yahoo.co.uk

**Keywords:** titanium alloys, obtaining, TiMoZrTaSi system, osseointegration

## Abstract

Considering the future trends of biomaterials, current studies are focused on the corrosion resistance and the mechanical properties of new materials that need to be considered in the process of strengthening alloys with additive non-toxic elements. Many kinds of titanium alloys with different biocompatible elements (Mo, Si, Zr, etc.,) have been recently developed for their similar properties with human bone. Four new different alloys were obtained and investigated regarding their microstructure, mechanical, chemical, and biological behavior (in vitro and in vivo evaluation), the alloys are as follows: Ti15Mo7Zr15Ta, Ti15Mo7Zr15Ta0.5Si, Ti15Mo7Zr15Ta0.75Si, and Ti15Mo7Zr15Ta1Si. There were changes with the addition of the silicon element such as the hardness and the modulus of elasticity increased. An MTT assay confirmed the in vitro cytocompatibility of the prepared alloys.

## 1. Introduction

In his study M. Niinomi affirmed that the biomaterial is a “synthetic material used to replace a part of a living system or to function in close connection with a living tissue” [1].

For the selection of biomaterials, in order to fabricate an implant, it is necessary to take into account a multitude of factors such as: economic, mechanical, electrical, environmental (chemical), safety (biological), thermal, surface, aesthetic, performance, and research [2,3].

To be a quality material and to maintain a long time in the human body without failures, it depends on certain factors such as material properties, design and biocompatibility of the material used, as well as other factors that are not under the direct control of the engineer, including in this category the technique used by the surgeon the patient, and the patient’s concerns [4,5].

Metals are some of the most widely used biomaterials for orthopedic implants, and more applications. They are known for their high wear resistance, ductility, and high hardness. The most commonly used metals for making implants are titanium and titanium alloys, stainless steels, and cobalt-chromium-molybdenum alloys. Titanium and its alloys are especially used to fabricate orthopedic implants due to the fact that its mechanical properties are similar to those of bone tissue [6,7]. 

In the human body the materials used as orthopedic implants interact with surrounding tissues by friction and wear while they are replacing a tissue or an organ. However, due to the fact that the human body is a corrosive environment, the corrosion is often accompanying the wear, and a corrosive wear is produced [8]. The corrosive wear produces a great amount of material loss, which is often greater than the corrosion induced amount and wear induced separately. Because of this, the implants alloys need to have both wear resistance and high corrosion properties. Studies on the frequently used metallic materials, such as the Ti-based alloys or Co-Cr-Mo systems, or 316 L stainless steel revealed important interactions of the corrosion with wear in the simulated physiological environment [9]. Such interactions of the biomedical implant materials with the human body influence the long-term use of the implant and potential performances of the metallic materials. 

Titanium biomaterials are highly sought for to their high resistance to breakage, stretching, and wear [10,11,12]. The combination of mechanical properties, corrosion resistance and biocompatibility recommend the titanium alloys for medical applications beside other conventional alloys (Co-Cr alloys, stainless steel) [13,14,15]. For example: 50% higher weight resistance ratio than stainless steel, 50% lower young modulus than stainless steel and Co-Cr alloys, etc. 

The Ti alloy recipe for biomedical devices is designed based on the effects of chemical elements on the microstructure and mechanical, and chemical and biocompatibility characteristics [16,17,18]. Among Ti alloys, monophasic β-Ti alloys are preferred for use as implant devices due to their low elasticity, comparable with that of bone (<30 GPa). This benefic characteristic is obtained by applying appropriate processing methods (cold working and solution heat treatment) [19,20]. 

The β-Ti phase stabilizing effect is obtained using either β-isomorphous stabilizers (V, Nb, Mo and Ta that have limited solubility in α-Ti phase) or β-eutectoid stabilizers (Fe, Mn, Cr, Ni, Cu, Si, H with solubility in both α-Ti and β-Ti phases) [21]. Zr and Sn have long been considered neutral in terms of β-Ti phase stabilization effects in Ti-based alloys. Other chemical elements such as Ti, Zr, Nb, Ru, Ta, Au, Mo and Sn are considered to have excellent biocompatibility, while a few other elements (Si, Fe, Mn) can studied to establish the effectiveness of use in the manufacture of implants. 

One of the promising alloying elements is Mo, especially for good biocompatibility, corrosion resistance and metallurgical effects, to which is added a low manufacturing cost comparatively with other alloying elements. Increasing the Mo content reduces the transition temperature of the β phase, decreases the elastic modulus and balances the pH in the body [22,23]. Recent research has shown that the use of about 18 wt.% Zr and 13 wt.% Mo as alloying elements for titanium has led to high values for yield stress and good ductility. The addition of Zr to the Ti-Mo alloy increases the stability of the beta phase and strengthens the solid solution [23]. Even though it increases the implant price, Ta is a beneficial alloying element in titanium, due the very good biocompatibility and decreasing the modulus of elasticity [15]. The addition of the cheap element Si (less than 1 wt.%), allows to decrease the modulus of elasticity of Ti15Mo alloy from above 44 GPa to 20 GPa [12].

The in vitro and in vivo studies proved the superiority of Ti alloys over C.P. Ti, the Ta-xZr alloys, determined the adhesion, and promoted the osteoprogenitor cells’ proliferation and differentiation. As a consequence, the alloys with a higher percentage of Ta showed better in vivo osteogenic activity so, the alloys Ta-Zr are a promising alternative in prosthetics [24]. The same, through adding the tantalum (Ta) and zirconium (Zr) the alloys, acquire a low elastic modulus and they exhibit the similar biomechanical characteristics to those of human bone [25,26,27]. In the same way, silicon (Si) is a chemical element which is found in varying amounts in any structure of the body, and is considered to be biocompatible with the body tissues. It is known that a higher dietary intake of silicon can lead to improve the bone density. Jugdaohsingh in 2007 [28] showed that silicon supplementation can cause the increasing in bone morphophysiological characteristics in the case of osteoporosis [28]. 

Four different alloys were obtained and investigated regarding microstructure, mechanical, chemical, and biological behavior (in vitro and in vivo evaluation), as follows: Ti15Mo7Zr15Ta, Ti15Mo7Zr15Ta0.5Si, Ti15Mo7Zr15Ta0.75Si, and Ti15Mo7Zr15Ta1Si.

## 2. Materials and Methods 

### 2.1. Material Preparation. Development of New TiMoZrTaSi Alloys

Titanium alloy obtaining technologies are still a challenge nowadays, due to the special protection conditions required and high costs of the metallurgical equipment. Due to a high affinity of titanium for oxygen, hydrogen, and nitrogen, during production of titanium-based alloys, special measures for protecting the melt metal against gases dissolving must be taken. These technological issues can be overcome by using suitable metallurgical equipment that allows for processing in a vacuum or in a controlled argon atmosphere. 

Considering these aspects, a Vacuum Arc Remelting installation from the Laboratory of Obtaining and Refining of Metallic Materials, Faculty of Materials Science and Engineering of the Politehnica University of Bucharest (www.eramet.ro, accessed on 10 September 2021) was used to obtain the experimental titanium-based alloys. 

To obtain the biocompatible Ti-Mo-Ta-Zr-Si alloys high purity chemical elements (99.5%) were used. The following alloys were obtained: Ti15Mo7Zr15Ta, Ti15Mo7Zr15Ta0.5Si, Ti15Mo7Zr15Ta0.75Si, and Ti15Mo7Zr15Ta1Si. To analyze the effect of silicon in alloys, the content of the other alloying elements (molybdenum, zirconium and tantalum) was kept constant, then the Si content was increased from 0.5 to 0.75 and 1 wt.%, respectively. The value of titanium content in alloys results from the balance equation.

The individual quantity of each element used for the production of Ti-based alloys mini ingots, weighing about 50 g, was established by calculation, considering the desired recipe.

The values of technological parameters of the obtaining process were the following: melting power of min. 55 kVA; melting current of min. 650 A, 60% DS, three-phase voltage; the vacuum level obtained with preliminary and diffusion vacuum pumps of 4.5 × 10^−3^ mbar; the inert gas supply was argon 5.3. For each mini-ingot, five successive melts were performed, on each side, in order to obtain a compositional homogenization of the alloys.

### 2.2. Microstructural Characterization Methods

To highlight the proportions obtained between the pure chemical elements and investigation of morphological analysis, an investigation was necessary. An X-ray energy dispersion, (SEM)—Vega 2 LSH Tescan Brno, Czech Republic performed the compositional analysis and morphological structure.

For the optical microstructures the Zeiss Axio Imager A1 upright microscope with multiple objectives (Carl-Zeiss-Strasse, Oberkochen, Germany).

The XRD analysis were performed using an Xpert Pro MPD diffractometer, with a Cu X-ray tube (Kα 1.54056°) (Panalytical, Almelo, The Netherlands). The diffraction pattern allowed us to identify the specific phases of the samples, using the High Score Plus software (version 3.1). 

The preparation of the sample included: cutting the samples to precise dimensions in order to characterize them with the help of a diamond disc cutting machine Metacut 302 (Metkon, Bursa, Turkey), grinding and polishing them, and then attacking with a special reagent (10 mL HF, 5 mL HNO_3_, 85 mL H_2_O solution).

### 2.3. Mechanical Properties

The analysis of the aspects related to the microhardness of the alloys was performed by the Vickers method using a CV-400 DM Microdurimeter (Bowers Group, Camberley, UK), and the samples were tested with a force of 50 gf (HV 50). 

In the latest years, indentation methods have been developed and applied to determine the modulus of elasticity and hardness of metals, plastics, polymer films, etc. Depending on the analyzed field, the fingerprint tests allow for obtaining the viscoelastic-plastic properties at the macro-micro- or nano-metric scale. 

In the field of metallic materials, the characterization by indentation has a major advantage over the classical testing methods on standardized test pieces, namely, the testing can be performed directly on the finished parts. The equipment used to determine the characteristics by indentation was a Tribometer UMT 2 with a Rockwell diamond penetrator (Bruker, Campbell, CA, USA), angle to the indenter cone of 120° and a spherical tip of radius of 200 μm, applying a force of 5N. For a more accurate determination, three determinations were made for each alloy. After completing the work steps and recording them by the UMT 2 software, the curves (depth vs. force) of the TiMoZrTaSi alloys were plotted through the VIEWER program.

### 2.4. In Vitro Cytocompatibility Assessment 

For cytocompatibility tests, the following materials were used: Hank’s balanced salt solution ((HBSS) Sigma Aldrich) and Dulbecco’s modified eagle medium ((DMEM) Sigma Aldrich) were used in culturing media for the MTT tests, specific materials were used and an antibiotic cocktail was used a stable solution, filtered, sterile of penicillin-streptomycin-neomycin (PSN), especially prepared for biology tests; the medium was completed with fetal bovine serum ((FBS) Sigma Aldrich), serum which is sterile and used for cell culture.

The cytotoxicity tests were performed on all materials and two directions were considered: the cell viability by measuring the metabolic activity of cells in vitro, as well as the modification in cell morphology as a result of the interaction with the metallic alloys. Some degradation products from alloys, as well as impurities or catalysts can be released into the culture medium and induce a cytotoxic effect on cells, important attention was given to the composition of the alloys. The tests of cytotoxicity were performed by incubating the samples of TiMoZrTaSi alloys with two types of cells: primary Albino rabbit fibroblasts and human MG63 osteosarcoma cells (the last one being line cells), The MTT assessment was used to measure the cell metabolic activity and the cytotoxic effect.

#### 2.4.1. Dermal Fibroblasts for Cells Culture 

To isolate the Albino rabbit dermal fibroblasts, the explant explants technique was used because the method is suitable to isolate cells with a homogeneous population by using very small tissue fragments. The method is appropriate especially for cells of a fibroblastic type, as is described in the literature [29]. 

For cytotoxicity tests the cells used were fibroblasts from the passage with No. 5. Briefly, from the culture substrate was detached a sub-confluent cell monolayer by using the trypsinization method. A mixture of solutions was used: (0.025% trypsin and 0.02% EDTA. The cells were collected in culture medium: the result was cell suspension. 

To measure the number of cells and the cell viability, 100 µL of 0.3% trypan blue was added to the 100 µL of cell suspension and the mixture was homogenized with the same, at room temperature; a 2 mL Eppendorf tube was used. A Neubauer cell counting chamber (Merck, Darmstadt, Germany) was used to count the number of cells: the mixture of cells and trypan blue was added on the chamber and the cells in the outer squares of the chamber were counted (10x objective of the microscope). The cell viability was calculated as follows:(1)$$Viability\left(\%\right)=\frac{No\;of\;alive\;cells\;\left(uncolored\right)}{No\;of\;alive\;cells+No\;of\;dead\;cells\;\left(colored\right)}\times 100$$

#### 2.4.2. Cytotoxicity Experiment and Cell Viability Evaluation

The culture media based on DMEM with 10% FBS and a 1% mixture of drugs (PNS) was prepared and the cells were cultured in an atmosphere of 5% of CO_2_, which was created at 37 °C and humidity of 97%, appropriate conditions for cell growing. The cells reached a status of sub-confluence after 4 days, when a mixture of enzyme trypsin with a concentration of 0.25% and 1 mM ethylenediamine tetraacetic acid (EDTA) was included. The resulted cells (fibroblasts) were cultured on 48-well plates, using a concentration of 5 × 10^3^ cells/well. 

The materials were sterilized by immersion in 70% absolute ethyl alcohol solution (sterile solution), for 30 min and successive washing with bi-distilled water and saline phosphate solution (especially reactive for cell cultures). The sterile samples were immersed 2 mL of complete culture medium (DMEM with addition of 10% fetal calf serum and 1% mixture of drugs, penicillin, streptomycin, and neomycin). The alloys and medium were incubated at 37 °C, 5% atmospheric CO_2_ concentration and 97% humidity, for 24 h.

The sterile alloy materials were deposited on the confluent cells layers and they were maintained in contact for various periods (24, 48 and 72 h). Finally, the MTT assay was performed in the aim to evaluate the cell viability. The cell viability was calculated with the relation:
Cell viability = (A_s_ × 100)/A_c_ [%](2)
where A_s_ is the absorbance after the MTT inclusion in the cultured cells with metallic alloy and A_c_ is the control absorbance (cells alone). The experiments were performed in triplicate for each tested alloy and the average value was considered.

#### 2.4.3. Cells Morphology

The cell viability after contact with the metallic alloys was analyzed by phase contrast optical microscopy (Olympus, Hamburg, Germany) compared with cells from control cultures.

A staining method using calcein AM was used to analyze the viable cell morphology. The cell membrane is permeable for calcein (calcemide), a non-fluorescent and colorless reactive. Calcein penetrates into the cell, and inside of the cell is hydrolyzed to a fluorescent green product from living cells, by reaction of the cytoplasmic esterase. The hydrolyzed calcein is detected for its fluorescence by using fluorescence microscopy; a Leica DM5500Q fluorescent microscope (Leica Microsystems, Heppenheim, Germany) equipped with an excitation filter of 455 nm and an emission of 530 nm was used.

### 2.5. In Vivo Biocompatibility Assessment

To highlight the biological interaction between alloys and the body tissues, each alloy was implanted in the compacta of the tibial crest in mature rabbits from the *Orychtolagus cuniculus* species, aged of 8–10 months, and a stainless-steel orthopedic rod was implanted in the control one. The experiment lasted 60 days and the experimental protocol was in accord to the European [30] and national legislation [31] in the matter of experimentation on animals using of a minimum number of animals for the proposed purposes and under the protocols of ISO 10993-1:2018 [32]. The prophylaxis, animal maintenance, the intra- and post-surgery maneuvers and euthanasia of animals for collecting the biological sampling were approved by the professional ethics committee of the Faculty of Veterinary Medicine from Iasi [30,31]. Alloys were implanted in the tranquilized animals with Xylazine/Ketamine and, an incision of about 0.5 cm was made on the tibial ridge and, depending on the size of each implant, a bone gap of about 0.3–0.5 mm in width and 0.4–0.8 mm in depth was made (surgical canal) which is the orthopedic electric drill, in which were inserted an orthopedic rod in the control and the implants in the four experimental rabbits. After them fixation, the periosteum and the regional fascia were sewn, followed by suturing the skin. To prevent the inflammatory processes, each animal was administered antibiotics for 6 days. Immediately after surgery and at the end of the experiment, radiological examinations were performed to highlight the position of the implants or some aspects related to the surrounding tissues, the radiographs were performed with the Intermedical Basic 4006 device (Intermedical S.R.L. IMD Group, Grassobbio, Italy). After the recording of the images using the X-CR Smart Examion digital system (Examion), they were saved in DICOM format and being read in the Radiant program. The CT scan evaluation was performed using the General Electric LightSpeed 16(General Electric Company, Milwaukee, Wisconsin, USA). The Hounsfield unit scale (UH) was used to identify and compare the radiodensity of the regional structures.

After collecting the implanted parts of tibia from each rabbit, they were fixed for 24 h in 10% formol solution, and during 6 weeks included into 15% EDTA, pH of 7 to be decalcified. After that, the slices were dehydrated using ethanol in decreasing concentrations followed by clarification with xylene and inclusion in paraffin. From each paraffin block, 10 microscope slides were cut at 5 μm each with the microtome SLEE cut 6062(SLEE, Mainz, Germany)., after that they were stained with the standard hematoxilin-eosin protocol and read under the microscope Olympus CX41(Olympus, Hamburg, Germany).

The anti-osteopontin (GmbH Aachen Germany, antibodies, ABIN2774904), anti-MMP2, and anti –MMP9 (Santa Cruz, Biotechnology (C-20): SC-6840) were used for obtaining the immunohistochemical staining. During 10 min, the slides were dewaxed and microwaved at 95 °C in 10 mmol citrate acid buffer pH6 followed by cooling them for 20 min and by twice washing in PBS (phosphate-buffered saline) for 5 min. After that, the slides were incubated overnight at 4 °C and in humid environment with primary antibodies, diluted 1:100. The next day, each slide was washed three times for 5 min in PBS then were incubated with the secondary antibodies.

To highlight the osteopontin (OPN) expression in cells, the goat anti-rabbit IgG secondary antibody was used, and the goat anti-mouse IgG secondary antibody showed the expression of MMP2 and MMP9. The microscopic sections were developed with 3,3′-diaminobenzidine (DAB) and with hematoxylin were finally counter stained. 

## 3. Results and Discussions

### 3.1. Composition and Microstructure of Ti-Based Obtained Alloys

The elemental chemical composition of the Ti-15Mo-7Zr-15Ta-xSi (0.5, 0.75, 1 wt.%) was measured by an energy dispersive X-ray spectroscopy analysis (EDX). To obtain a precise value, the average was performed from the ten determinations, from five different areas of the samples. Table 1 presents the average results of ten EDX points of the studied alloys. The composition of the alloys was uniform and no molten metal inclusions were found.

In Figure 1 is highlighted the structure of Ti-15Mo-7Zr-15Ta-xSi (0.5, 0.75, 1 wt.%) by optical microscopy. The structure of these alloys is directly influenced by the amount of α or β stabilizing elements added. Titanium has a number of features that distinguish it from the other light metals; at 882 °C (1620 °F) titanium undergoes an allotropic transformation from a low-temperature, hexagonal close-packed structure (hcp) (α) to a body-centered cubic (β) phase that remains stable up to the melting point.

The influence of beta stabilizers (molybdenum, tantalum, and silicon) can be observed by the β phase between the intragranular primary α phase precipitations. The grain size, typically ranging from 0.5 to 5 mm (0.02 to 0.2 in.), develops during cooling through the β phase field, with slow cooling rates resulting in larger β grains. The formation of dendrites specific to titanium alloys indicates that casting solidification has very rapid cooling in the copper mold.

The solidification understanding and microstructural formation at titanium alloys is more difficult than many alloys because of the complicated phase transitions that occur.

Beta stabilizing elements such as Mo, Si, and Ta favored the presence of the β phase in experimental alloys, the alloys having a cubic structure with centered volume presented in diffraction patters from Figure 2.

Using the Mo-Si, Mo-Ti, and Ti-Ta binary diagrams, the following three stable compounds were determined in these alloys: β-Ti (reference code: 01-089-4913), tantalum Titanium (Reference code: 00-052-0960), and molybdenum titanium (Reference code: 03-065-9395). These compounds that were formed in alloys were identified by XRD analysis. The main characteristics for β-Ti are as follows: crystal system; Cubic, space group; Im-3m; a (Å): 32,830; b (Å): 32,830; c (Å): 32,830; Alpha (°): 900,000; Beta (°): 900,000; Gamma (°): 900,000; calculated density (g/cm^3^): 4.49; and volume of cell (10^6^ pm^3^): 35.38. The main characteristics for TaTi are as follows: crystal system: orthorhombic, space group: Cmcm; a (Å): 30,380; b (Å): 49,570; c (Å): 46,860; Alpha (°): 900,000; Beta (°): 900,000; Gamma (°): 900,000; calculated density (g/cm^3^): 6.37; and volume of cell (10^6^ pm^3^): 70.57. The main characteristics for MoTi are: crystal system; Cubic, space group; Im-3m; a (Å): 31,750; b (Å): 31,750; c (Å): 31,750; Alpha (°): 900,000; Beta (°): 900,000; Gamma (°): 900,000; and volume of cell (10^6^ pm^3^): 32.01.

### 3.2. Mechanical Properties of Ti-Mo-Zr-Ta-Si Alloys

Hardness measurements results are presented in Table 2 of the investigated Ti15Mo7Zr15TaxSi (x = 0.5, 0.75, 1%) alloys. Values were between 237 HV for Ti15Mo7Zr15Ta and 362 HV for the Ti15Mo7Zr15Ta1Si. With the increase in the percentage of silicon, the value of hardness also increased.

Figure 3 shows the response of the alloys during an indentation test, the force-displacement curve. In Table 3 are presented the alloys results of the indention test. The force applied to the indenter increases continuously in the type of loading phase, is kept constant at the maximum value during the holding phase, and decreases to zero in the unloading phase. Depending on the alloys studied, the maintenance phase may or may not be present. 

The holding phase is also called creep, the indentation depth varies over time, the force being constant. The maximum depth is obtained for the maximum value of the applied force. Due to the plastic deformations, after the complete discharge, a residual, permanent depth is obtained.

The values obtained were in the range 23 GPa (Ti15Mo7Zr15Ta)–71 GPa (Ti15Mo7Zr15Ta1Si). From the indentation results, it can be seen that the modulus of elasticity increases with the increase in the percentage of Si.

### 3.3. In Vitro Cytotoxicity of Ti-Mo-Zr-Ta-Si Alloys

The materials tested as components in bone prosthesis must be tested for their in vitro and in vivo biocompatibility before using with a human body. Various methods were used for cytocompatibility studies and the MTT assay is a standardized one, either the contact direct method or extract method, depending on the nature of the material and application. The direct contact method is the standard method for alloys used as implants and it was applied in our study. Additionally, in the aim to analyze the material’s effect on cells a live/dead staining was performed and analyzed. Standard periods of time were used for incubation the cells with metallic materials: 24 h, 48 h, and 72 h of incubation and the cell culture was compared with a control one (cells without material). Figure 4 presents the results obtained for cell viability after calculation (and reported to a control cell culture). 

According to the Formula (1), the viability of the cells after calculation was found at 98.5%. In a culture vial with an area of 75 cm^2^ the total number of viable cells was found as 4.5 × 10^6^.

Data presented in Figure 4, revealed that all Ti-Mo-Zr-Ta-Si materials are cytocompatible; for all materials the cell viability has values over 80% from a standard cell culture (72 h of incubation). By adding the Si element in alloys, the cells (fibroblasts) viability is slightly modified by the content of this element, and the cell viability was higher by increasing the concentration of Si.

In the aim to evidence the cells morphology the Ti-Mo-Zr-Ta-Si materials were incubated with primary fibroblasts isolated from albino rabbit dermis and MG63 cells for 72 h and then stained (Figure 5 and Figure 6).

After the contact with Ti-Mo-Zr-Ta-Si material (the dark spaces of each image) it can be concluded that a cell monolayer is formed, which is not different from the growth control. The cells are elongated, which is characteristic for the tested ones (fibroblasts). The remained dark areas are associated with the space occupied by the material; it can induce some mechanical actions which affect the uniformity of the monolayer. However, the density or morphology of the cells placed at the border with Ti-Mo-Zr-Ta-Si materials was not observed, indicating no effect of these alloys on cells.

An increased cell growth density was observed for MG63 osteoblasts cells by comparison with fibroblasts culture. It can be explained by a higher growth rate of the line of osteoblasts MG 63. The fibroblasts used were primary cells, with a slower tendency to grow. Analyzing the fluorescence microscopy data obtained by contacting Ti-Mo-Zr-Ta-Si materials with fibroblasts and osteoblasts, no modification in growth and morphology can be observed. After contact with metallic alloys the cells have the characteristics compared with the control culture [33,34,35].

### 3.4. In Vivo Biocompatibility of Ti-Mo-Zr-Ta-Si Alloys

An X-Ray performed in the region of the femur-tibio-patellar joint in latero-lateral incidence identified the presence of implants and in the adjacent peri-implant tissues no abnormal radiological changes were identified, both in the control and in the implanted rabbits (Figure 7).

### 3.5. Histological Analysis of the Biocompatibility of the Studied Implants

Because each individual has a slightly different bone density, even if they are females and similar in ages and weight, the bone radiodensity highlighted by CT scan and measured in UH slightly varies, falling within the physiological average of 1125 UH [36,37]. In the CT scan, the Hounsfield unit shows some values which are in direct relation to the degree of X-ray attenuation that is allocated to each pixel, finally obtaining the images which express the density of each tissue [38,39].

Regarding the radiodensity of the implant surrounding tissues, after 60 days of experiments, in the control rabbit, the newly formed tissue in the implantation gap measures 793 UH, which is determined by the placing of the iron rod into the bone marrow duct. In the case of TI15Mo7Ta15 the new bone counted a density of up to 698 UH (and its bone compacta between 1359 and 2127 UH), and about 633 UH (the tibial compacta about 1520 UH) in the case of the alloy Ti15Mo7Zr15Ta1.0Si. Lesser radiopaque tissue of up to 573 UH (at tibial compacta between 1222 and 2124 UH) was registered in the case of the Ti15Mo7Zr15Ta0.5Si alloy and the higher radio-opacity of approximately 755 UH (bone compact of about 2079 UH) in Ti15Mo7Zn15Ta0.75Si (Figure 8). The radio-opacity identified in the newly formed tissue around the implants is associated with the fibrous tissues with cartilage islands and ossification punctures formed during the bone remodeling and implants osseointegration, which is confirmed by histologic analyses [11,12,40]. 

In the case of the control and implanted rabbits, into the inner cambium there are present the mesenchymal cells which firstly differentiate in osteoprogenitor cells (Figure 9, first row of images), after that they are present in osteoblasts which are involved in osteoid production.

During surgery, all implanting ditches were covered by periosteum, since that the osteoprogenitor cambium cells extensively proliferate firstly by producing the cartilage then the new bone into the fracture callus [41,42]. During the remodeling process of the subperiosteal area, the mesenchymal cells that differentiate in osteoblasts maturate into osteocytes that have a role in redirecting the new bone trabeculae along the force lines of the osteonal system and integrate them with the compact bone (Figure 9-Middle row of images).

The space between the periosteum and the alloy is completed by mesenchymal stem cells that differentiate into the bone-forming osteoblasts and the chondrocytes which produce the cartilage. During endochondral ossification, the cartilage is used as a template for additional bone formation [42,43].

Titanium alloys are covered by a 3–10 nm passive layer which may attract the calcium and phosphate ions to induce some proteins (collagen type I, osteocalcin, osteonectin, sialoprotein, proteinolipids, fibronectin, and vitronectin) being able to bind with osteoblasts [36,44,45,46], firstly forming a fibrous capsule. This fibrous capsule was formed in our experiment too (Figure 9-Last row of images). It is non-vascularized and into its inner area were found the mesenchymal cells which follow the differentiation steps into osteoprogenitor cells and osteoblasts that are involved in the synthesis of new bone by intramembranous ossification. 

At the control rabbit, the surrounding capsule is thinner but dense. In the case of Ti15Mo7Zr15Ta, nearby the alloy is registered a thicker capsule and the agglomeration of the mesenchymal stem cells. The fibrous capsule which surrounds the Ti15Mo7Zr15Ta0.5Si alloy consists of low number of bone trabeculae inside the new bone, they well differentiate through intramembranous ossification in the case of Ti15Mo7Zr15Ta0.75Si or mixt, intramembranous, and endochondral in the case of Ti15Mo7Zr15Ta1Si alloy.

In all cases, just nearby the alloys, the fibrous capsule generates a strong contact both with the surrounding new bone and the alloy pores to such an extent that it forms a bony sleeve around it [47] which is included into the callus after 60 days post implant [11,12]. 

In control and implanted rabbits OPN was expressed in cells of the intertrabecular spaces, into the organic extracellular matrix and the same in osteocytes in the periosteum and in vicinity of the implants, showing an intense activity and differentiation of bone cells (Figure 10-First row of images).

Osteopontin (OPN) has been shown to have the potential to enhance the in vivo osseointegration [48,49]. It is involved in promoting of the cell adhesion through binding integrins which induce the intracellular signal transduction for promoting the proliferation and migration of the cells, etc., [50] for repairing the bone structures.

Some studies have shown that OPN can promote the secretion of MMP to increase the cell migration and cell adhesion by inducing the secretion and regulation of MMP-9 or MMP-2 in different cells. So, the bone cells which express the OPN, MMP-9, or MMP-2 are actively involved in the production and degradation of the extracellular matrix during bone remodeling [51,52]. 

In all cases of our study, the high MMP2 and MMP9 activity was shown in osteoprogenitor cells and osteoblasts from the intertrabecular spaces and in the osteocytes (Figure 10-The second and the third rows of images). 

## 4. Conclusions

In the design of new from the Ti-Mo-Zr-Ta-Si system, some important properties of new alloyed materials were taken into account: the influence of the chemical elements on the chemical and mechanical properties, and the level of biocompatibility with the body tissues and the corrosion resistance. Different batches from the Ti15Mo7Zr15TaxSi (x = 0.5%, 0.75%, 1%) system were performed. The obtaining efficiency for the experimental alloys was 98.46–99.9%. The chemical composition of the mini ingots obtained was very similar to the designed composition, which shows that the losses by evaporation or oxidation were very small (process efficiency was between 98% and 99%).

The effect of adding silicon to the Ti-Mo-Zr-Ta system was very well observed from the mechanical properties: the hardness and the modulus of elasticity increased.

In vitro cytocompatibility tests performed on fibroblasts and osteoblasts indicated no effect of the Ti-Mo-Zr-Ta-Si materials on cells and non-toxic interactions. A slight increase in cell viability was observed at higher content of Si in alloys. The staining of cells and fluorescence microscopy does not indicate modifications in growth and morphology of both types of cells after interaction with the new alloys analyzed. 

The in vivo investigation spotlighted the ability of the titanium alloys containing Mo, Zr, Ta and Si to promote a higher osseointegration for 60 days after implantation. The use of metals in different concentrations in the proposed alloys does not influence their biocompatibility with the bone tissue. The high positively for OPN, MMP2, and MMP9 of osteoprogenitor cells, osteoblasts, and osteocytes in the peri-implant areas shows intense bone remodeling activity with osseointegration of the alloys.

## Figures and Tables

**Figure 1 materials-14-06806-f001:**
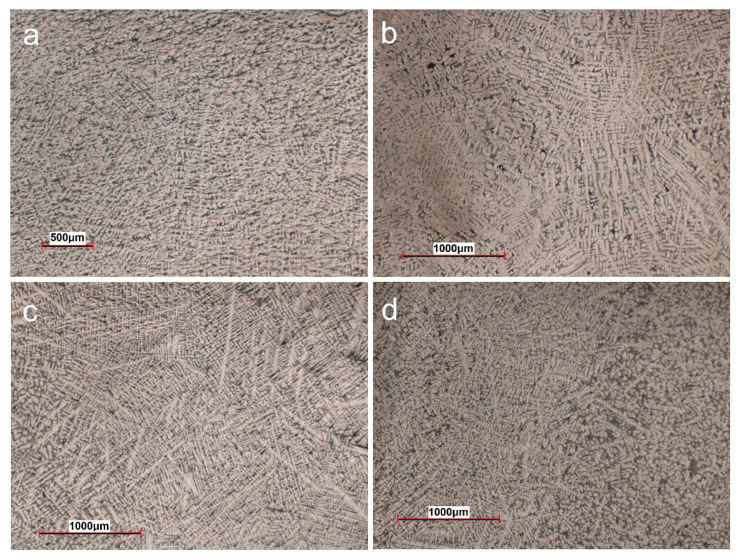
Optical microscopy images (50× magnification): (**a**) Ti15Mo7Zr15Ta, (**b**) Ti15Mo7Zr15Ta0.5Si, (**c**) Ti15Mo7Zr15Ta0.75Si, (**d**) Ti15Mo7Zr15Ta1Si.

**Figure 2 materials-14-06806-f002:**
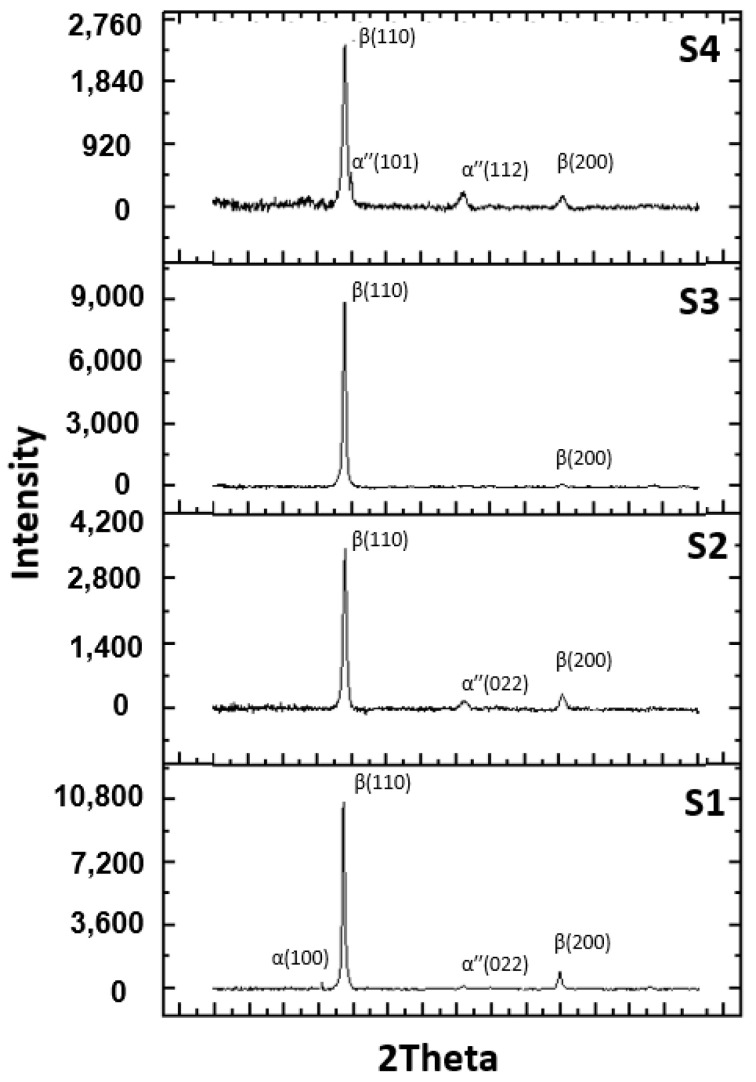
Diffractograms of experimental alloys: **S1**, Ti15Mo7Zr15Ta, **S2**, Ti15Mo7Zr15Ta0.5Si, **S3**, Ti15Mo7Zr15Ta0.75Si, **S4**, Ti15Mo7Zr15Ta1Si.

**Figure 3 materials-14-06806-f003:**
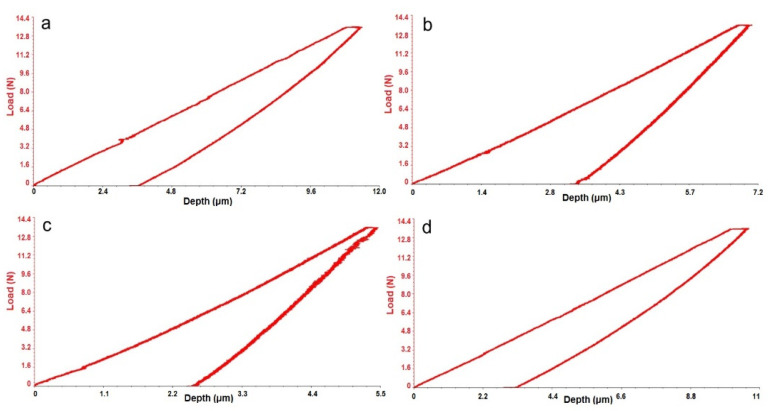
Micro-indentation graphic: (**a**) Ti15Mo7Zr15Ta, (**b**) Ti15Mo7Zr15Ta0.5Si, (**c**) Ti15Mo7Zr15Ta0.75Si, (**d**) Ti15Mo7Zr15Ta1Si.

**Figure 4 materials-14-06806-f004:**
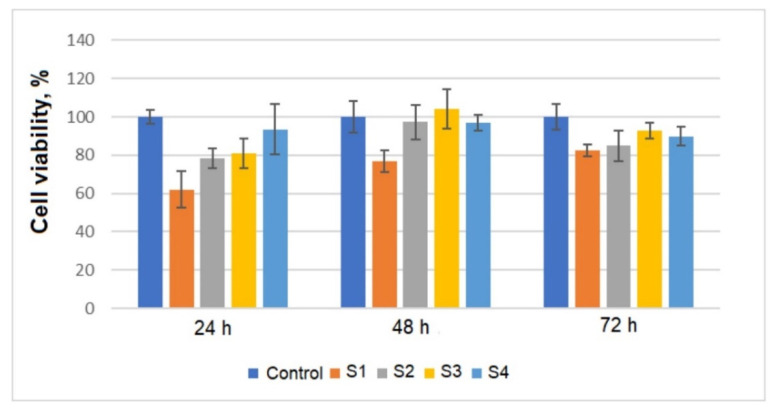
Cell viability data after contact of the cells with Ti-Mo-Zr-Ta-Si metallic alloys (24 h, 48 h and 72 h).

**Figure 5 materials-14-06806-f005:**
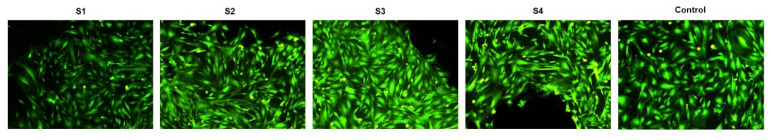
Fluorescent microscopy (staining with calcein AM) for cells (Albino rabbit fibroblasts) at 72 h cell culture with Ti-Mo-Zr-Ta-Si materials (10× objective magnification).

**Figure 6 materials-14-06806-f006:**
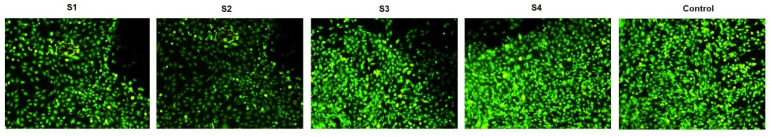
Fluorescent microscopy (staining with calcein AM) for cells (MG63) at 72 h cell culture with Ti-Mo-Zr-Ta-Si materials (10× objective magnification).

**Figure 7 materials-14-06806-f007:**
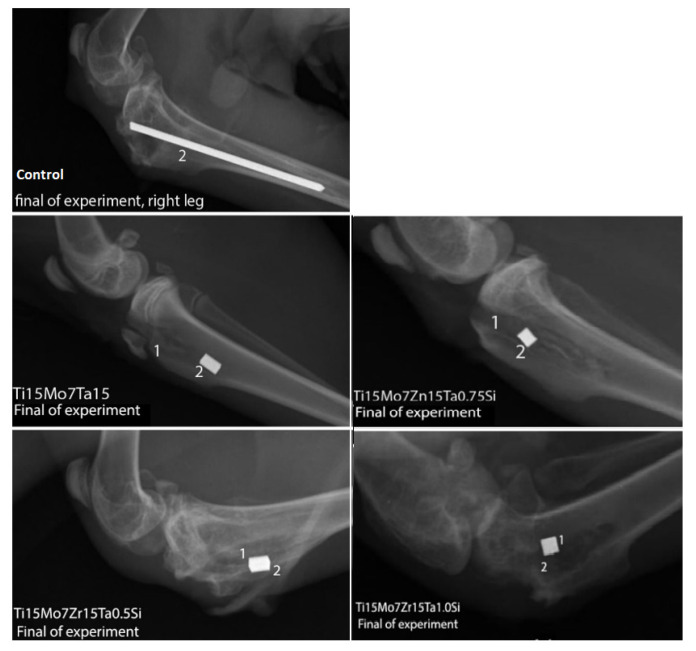
X-Ray in control and experimental rabbits, 1, alloy; 2, implantory breach.

**Figure 8 materials-14-06806-f008:**
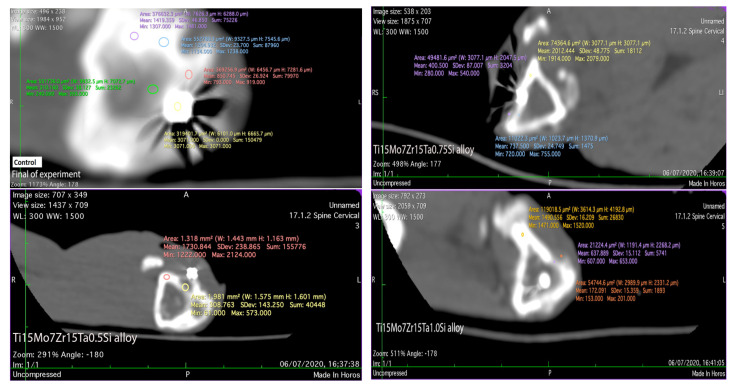
CT scan of the implanted area in control and experimental rabbits after 60 days.

**Figure 9 materials-14-06806-f009:**
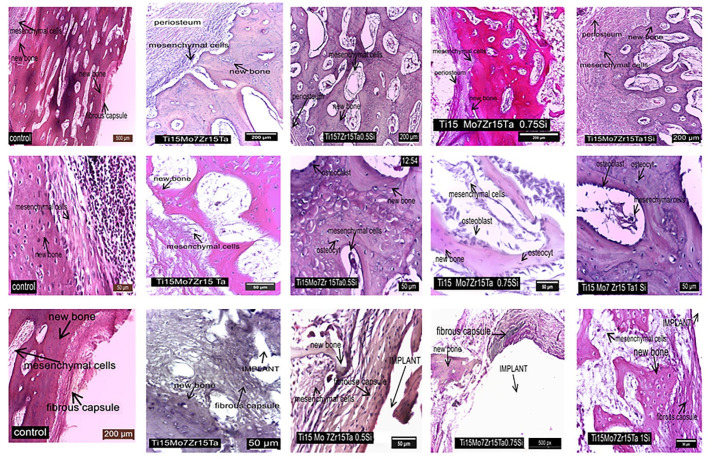
Some aspects about the peri-implant areas in control and experimental rabbits, stain HE. First row of images: the periosteum nearby the implant, in control and implanted rabbits; the middle row of images: some aspects concerning osteogenesis of new bone between periosteum and alloys in control and implanted rabbits; the last row of images: the peri-implant aspects of the fibrous capsule structure and the new bone maturation in control and implanted rabbits.

**Figure 10 materials-14-06806-f010:**
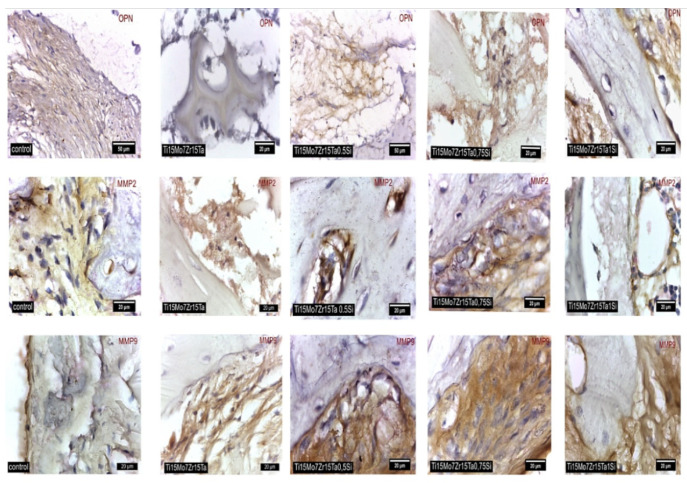
Stain for OPN and MMP2 and MMP9 in control and experimental rabbits nearby the alloys. First row of images: stain IHC, positively cells for osteopontin (OPN); second row of images: stain ICH, positively cells for metalloproteinases 2 (MMP2); third row of images: stain ICH, positively cells for metalloproteinases 9 (MMP9).

**Table 1 materials-14-06806-t001:** Average elemental compositions obtained for Ti-15Mo-7Zr-15Ta-xSi (0.5, 0.75, 1 wt.%) obtained.

Sample		S1 Ti15Mo7Zr15Ta	S2 Ti15Mo7Zr15Ta0.5Si	S3 Ti15Mo7Zr15Ta0.75Si	S4 Ti15Mo7Zr15Ta1Si
Average chemical composition	Ti (wt.%)	73.85 ± 0.3	66.50 ± 0.1	66.20 ± 0.1	73.00 ± 0.1
	(at.%)	86.41 ± 0.3	77.05 ± 0.1	77.73 ± 0.1	80.82 ± 0.1
	Mo (wt.%)	9.00 ± 0.1	11.00 ± 0.2	9.00 ± 0.3	10.00 ± 0.2
	(at.%)	3.18 ± 0.1	6.44 ± 0.1	4.64 ± 0.3	7.18 ± 0.2
	Zr (wt.%)	7.15 ± 0.2	7.00 ± 0.3	7.00 ± 0.1	8.00 ± 0.1
	(at.%)	5.71 ± 0.2	4.51 ± 0.3	5.23 ± 0.1	5.63 ± 0.1
	Ta (wt.%)	10.00 ± 0.2	15.00 ± 0.1	17.00 ± 0.3	8.00 ± 0.2
	(at.%)	4.68 ± 0.2	10.72 ± 0.1	11.61 ± 0.3	5.81 ± 0.2
	Si (wt.%)	-	0.50 ± 0.3	0.80 ± 0.3	1.00 ± 0.3
	(at.%)	-	1.28 ± 0.3	0.76 ± 0.3	0.99 ± 0.3

**Table 2 materials-14-06806-t002:** The hardness values of experimental alloys.

Alloys	Ti15Mo7Zr15Ta	Ti15Mo7Zr15Ta0.5Si	Ti15Mo7Zr15Ta0.75Si	Ti15Mo7Zr15Ta1Si
HV	237.27 ± 2.4	339.03 ± 3.4	357.15 ± 3.3	362.83 ± 3.5

**Table 3 materials-14-06806-t003:** Micro-indentation results.

Sample	Loading Deformation [N]	Release Deformation [μm]	Young Modulus [GPa]	Stiffness [N/μm]	Specimen Poisson Ration
S1	13.52 ± 0.3	11.40 ± 0.2	23.17 ± 0.4	2.04 ± 0.1	0.23
S2	13.54 ± 0.1	7.07 ± 0.2	51.79 ± 0.3	4.52 ± 0.1	0.23
S3	13.54 ± 0.9	5.73 ± 0.5	66.24 ± 0.2	5.12 ± 0.2	0.23
S4	13.53 ± 0.1	5.57 ± 0.5	71.13 ± 0.3	5.38 ± 0.1	0.23

Five determinations were carried out in different areas of the samples with dimensions of 10 mm × 10 mm × 5 mm.

## Data Availability

Not applicable.

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
