# Peer review of "Design, Synthesis, and Preliminary Evaluation for Ti-Mo-Zr-Ta-Si Alloys for Potential Implant Applications"

_materials, 2021, doi:10.3390/ma14226806_

Round 1

Reviewer 1 Report

The authors should provide the composition of the alloys by EDS in at. % in Table 2. 

There is no need for Table 1, as that information is unnecessary. The designation of samples (S1, S2,...) can be included in Table 2.

Why did the authors denote samples as R1, R2,... in Fig. 2, while they have used S1, S2,... in other places?

What is the grain size of each alloy?

Ref. to Fig. 2: (i) That's not how XRD patterns are shown. Please use the standard way so that the peak shift, broadening, and peak intensity can be clearly identified. (ii) What is the volume fraction of both phases from the XRD patterns and what is the reason for the phase transformation?

The hardness values of all the alloys in Table 3 are quite similar. (i) What is the reason for that? (ii) What is the error bar?

Why S1 alloy has extremely low Young's modulus?

The authors should show the XRD patterns of alloys and microscopy images after the mechanical analysis?

The paper is riddled with spelling and grammatical mistakes. The author should address this issue to improve the quality of the paper.

Author Response

Please see file attached!

Reviewer 2 Report

Comments and suggestions to the authors are contained in the attached file. 

Round 2

Reviewer 1 Report

The authors have addressed my comments and I recommend the acceptance of this paper.